# Microscale Schottky superlubric generator with high direct-current density and ultralong life

Xuanyu Huang [1,2,3], Xiaojian Xiang[1,4,5], Jinhui Nie[1,4,5], Deli Peng[1,4], Fuwei Yang [1,4], Zhanghui Wu [1,4], Haiyang Jiang[1,5], Zhiping Xu [4] & Quanshui Zheng [1,2,3,4,5✉]

Miniaturized or microscale generators that can effectively convert weak and random mechanical energy into electricity have significant potential to provide solutions for the power supply problem of distributed devices. However, owing to the common occurrence of friction and wear, all such generators developed so far have failed to simultaneously achieve sufficiently high current density and sufficiently long lifetime, which are crucial for real-world applications. To address this issue, we invent a microscale Schottky superlubric generator (S-SLG), such that the sliding contact between microsized graphite flakes and n-type silicon is in a structural superlubric state (an ultra-low friction and wearless state). The S-SLG not only generates high current ($\sim$210 Am$^{-2}$) and power ($\sim$7 Wm$^{-2}$) densities, but also achieves a long lifetime of at least 5,000 cycles, while maintaining stable high electrical current density ($\sim$119 Am$^{-2}$). No current decay and wear are observed during the experiment, indicating that the actual persistence of the S-SLG is enduring or virtually unlimited. By excluding the mechanism of friction-induced excitation in the S-SLG, we further demonstrate an electronic drift process during relative sliding using a quasi-static semiconductor finite element simulation. Our work may guide and accelerate the future use of S-SLGs in real-world applications.

[1] Center for Nano and Micro Mechanics, Tsinghua University, Beijing 100084, China. [2] Department of Mechanical Engineering, Tsinghua University, Beijing 100084, China. [3] State Key Lab of Tribology, Tsinghua University, Beijing 10084, China. [4] Department of Engineering Mechanics, Tsinghua University, Beijing 100084, China. [5] Institute of Superlubricity Technology, Research Institute of Tsinghua University in Shenzhen, Shenzhen 518057, China. ✉email: zhengqs@tsinghua.edu.cn

With the rapid development of nanotechnology and microfabrication technology, miniaturized sensors and devices are constantly emerging in a vast number of applications in the fields of Internet of Things, sensor networks, big data, personal health systems, and artificial intelligence[1–5]. To date, these sensors and devices have mostly been powered by line cords, or by batteries and external chargers, and thus have had limited applications[6,7] particularly in independent, sustainable, maintenance-free operations of implantable biosensors, remote and mobile environmental sensors, nano-/micro-scale robots, and portable/wearable personal electronics[7–10]. For continued use, these devices require small, wireless, portable, and sustainable power supplies.

As a promising solution to these challenges, nanogenerators were first proposed in 2006[1] to convert weak and random mechanical energy into electricity. Since then, many types of nanogenerators have been proposed, such as piezoelectric nanogenerators (PENGs)[1,11], triboelectric nanogenerators (TENGs)[12], and electret-based microgenerators (EBMGs)[13]. More recently, the principle of generating direct current based on relatively sliding Schottky junctions is proposed[3,14–19], namely Schottky generators (S-Gs) was proposed. Compared with previously proposed nanogenerators, S-Gs have a simpler structure, and some show higher current densities[3,15,20]. However, to the best of our knowledge, all S-G reported so far have failed to simultaneously achieve sufficiently high current density and sufficiently long lifetime for real-world applications[3,15]. The highest current densities of the reported S-Gs were in the range of $10^4$–$10^7$ Am$^{-2}$, which were generated through the friction of nanotips on semiconductor substrates[3,20,21] with respect to a very high normal pressure of $1$–$10$ GPa. The corresponding currents decayed completely in the first 1–5 cycles[3,21]. In contrast, the highest lifetimes of reported S-Gs were in the range of 3,600–10,000 cycles[15,18]. The high lifetimes are credited to the surface-to-surface contact sliding between the macroscopic MoS$_2$ or graphene and the semiconductor with respect to a low normal pressure of 0.05–5 MPa, which the generated current densities were exceedingly low, around 0.033–1 Am$^{-2}$ [15,18]. Even though in these low-friction S-Gs, notable wear was observed[15,18].

The fundamental challenge of most reported S-Gs is rooted in the mechanism of generating current from friction-induced excitation[3,19,22,23]. However, there may exist another mechanism of generating current in a sliding Schottky joint, namely the mechanism of depletion layer establishment and destruction (DLED)[15,16,24,25]. This fundamental challenge can be solved and the DLED mechanism can be validated using a specific Schottky joint, which is in a state of structural superlubricity (SSL)—a state of ultralow friction and wearless between two solid surfaces[26]. Since the first realization of microscale SSL in the atmospheric environment in 2012[27], in addition to the realizations of high-speed SSL ($25$–$293$ m/s)[28,29], SSL has attracted wide-ranging interest in academic studies and practical applications for obtaining a revolutionary solution for friction and wear probems[26]. For example, as an application of SSL, several types of superlubric generators (SLGs) based on capacitors, triboelectric or electrets have been proposed[30]. SLGs are designed to achieve the maximum electric current density allowed by the dielectric material, which is three orders of magnitude higher than the maximum density of all reported TENGs[31] and PENGs[32], with nearly 100% conversion efficiency and long lifetime. In this study, we demonstrate a Schottky superlubric generator (S-SLG) as the physical prototype of superlubric generators that can not only generate a stable and high current density of ~210 Am$^{-2}$ and power density of ~7 Wm$^{-2}$, but more importantly, achieve a long lifetime of at least 5,000 cycles while maintaining stable high electrical current density (~119 Am$^{-2}$). By excluding the mechanism of friction-induced excitation in our

S-SLG, it is revealed that there must be other mechanism(s) of generating currents in Schottky generators. We further demonstrate through finite element simulations that DLED is the most likely mechanism of generating current in S-SLGs.

## Results

**Structure of Schottky superlubric generator (S-SLG).** The S-SLG prototype was designed in three steps. In the first step, we selected a graphite flake that was cleaved by shear from a square graphite mesa of $4 \times 4 \times 2.6\,\mu$m with a self-retracting motion (SRM) property[33] (see the Methods section and Supplementary Section 2.1 for details). The graphite mesa was made of highly ordered pyrolytic graphite (HOPG) with an Au film of 100 nm thickness (fabrication details are provided in Supplementary Section 1 and the Methods section). The cleaved surface (or bottom surface) of an SRM flake is a single-crystalline graphene sheet and is superlubric[34].

In the second step, we transferred the graphite flake onto an atomic smooth surface of n-type silicon with a doping concentration of $N_D = 10^{15}$ cm$^{-3}$ (see the Methods section and Supplementary Section 2.1 for details), which was coated with an Al electrode on the back of the film (Fig. 1a; fabrication process is presented in the Methods section). The measured work functions of the graphite flake and n-type silicon (n-Si) film are 4.77–4.82 eV and 4.36–4.42 eV respectively, as detailed in the Supplementary Section 2.2 and Supplementary Fig. 3. Owing to the difference in work functions, electrons are automatically transferred between the graphite and n-Si contacted surfaces to form a depletion layer and a Schottky barrier at equilibrium (Supplementary Fig. 9a shows the current–voltage ($I$–$V$) characteristic curve).

In the third step, we pressed on the top of the graphite flake using a conductive atomic force microscope (AFM) tip (coated with Au) to induce sliding between the superlubric graphite surface and the smooth n-Si surface through lateral motion of the piezoelectric displacement platform; Fig. 1b shows the optical image of the experimental setup. A current measurement circuit was connected to the conductive AFM tip and the Al electrode coated on the bottom of the n-Si. When the graphite flakes were relatively sliding against the n-Si, we measured the current and frictional force using the external circuits and lateral force mode of the AFM, respectively. The positive direction of the current measurement was from the graphite flake to n-Si.

The measured current is plotted in Fig. 1c for the first 2,000 cycles as we slid the graphite flake with a displacement amplitude of 2 μm and speed of 4 μm/s under a controlled normal force of $F_N = 22.3\,\mu$N applied by the tip upon the graphite flake (see Methods, Supplementary Section 3.1, and Supplementary Movie 1 for more details). The results show that the current during each sliding cycle remains nearly constant and gradually increases with the number of sliding cycles; noise current measurements shown in Supplementary Fig. 5 and the decay and disappearance of the current after the graphite flake layered, as shown in Supplementary Fig. 6, confirms that the current is caused by the sliding of the graphite/n-Si interface instead of the graphite/graphite interface. We then performed a longer (~5,000 cycles) sliding test with the same normal force but increased the sliding speed from 4 to 24 μm/s. Figure 1d shows that the current (blue line) $I$ increases with the number of sliding cycles under different sliding speeds; the maximum current reached 1.9 nA; and the maximum current density was 119 Am$^{-2}$ for the contact area of $4 \times 4$ μm, which is similar to the maximum possible current density (~150 Am$^{-2}$) at a sliding speed 1 mm/s for capacitor-based SLGs before electric breaking[30]. In addition, the measured friction force (red line) declines[35] in the first 64 cycles and is stabilized in the subsequent

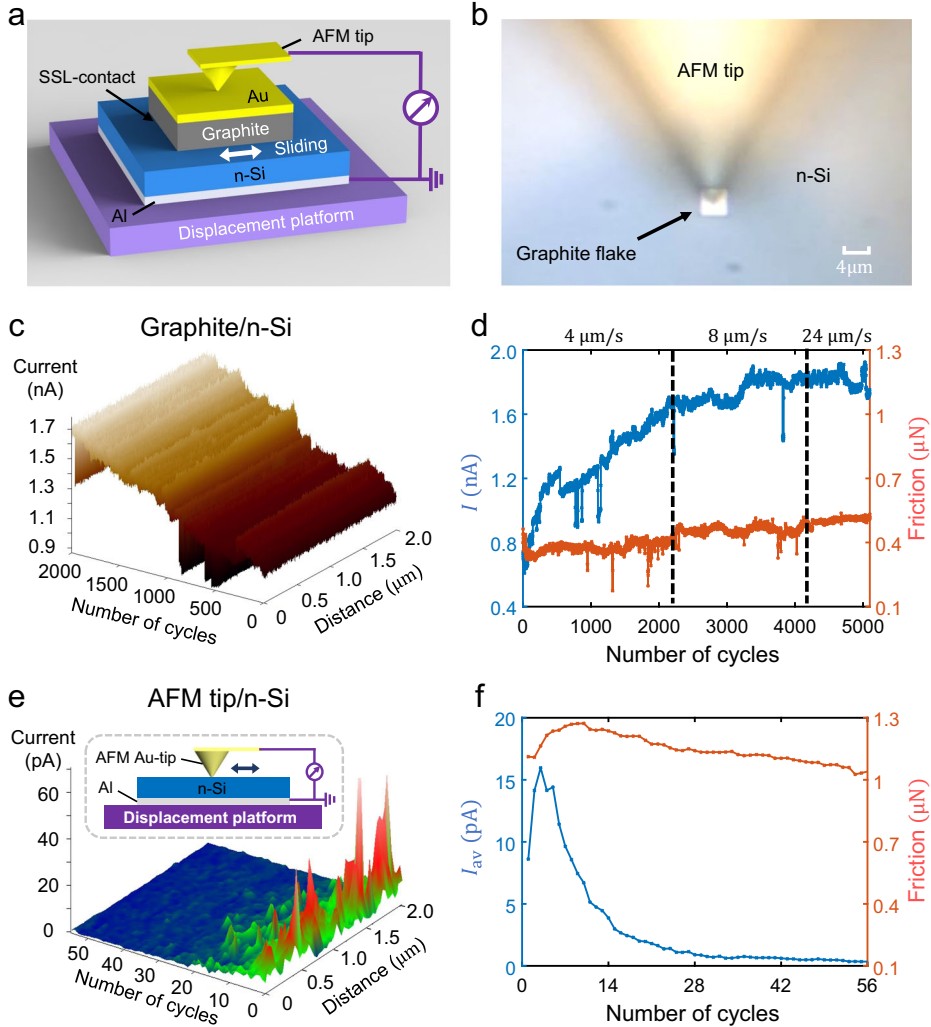

**Fig. 1 Output current and friction measurement of Schottky superlubric generator (S-SLG) and comparison with an ordinary Schottky generator (S-G).** **a** Structure of a graphite/n-Si S-SLG. **b** Optical microscopic image of a graphite/n-Si S-SLG. **c** Current maps in the first 2,000 cycles of a graphite/n-Si S-SLG with speed of 4 μm/s under a controlled normal force of $F_N = 22.3$ μN. **d** Relationship between friction (red) and current $I$ (blue) with sliding cycles of a graphite/n-Si S-SLG at different speeds. **e** Current maps in the first 56 cycles of an atomic force microscope (AFM) tip/n-Si ordinary S-G, and an illustration of the experimental setup. **f** Relationship between friction (red) and average current $I_{av}$ (blue) with sliding cycles of an AFM tip/n-Si ordinary S-G.

thousands of slides at different sliding speeds (only slowly increasing with sliding speed). As discussed in detail later, the mechanism of generating electricity in the S-SLG is most likely not the friction-induced excitation mechanism; rather, it is more likely to be DLED[15,36].

**Comparation between S-SLG and AFM tip/n-Si ordinary S-G.** For comparison, we performed an experiment by sliding an AFM Au-tip with a radius of curvature of 35 nm and normal force 4.49 μN over the same n-Si film. The measured current versus sliding cycles is plotted in Fig. 1e, with the inset indicating the experimental setup. The results show that the AFM tip/n-Si ordinary S-G produces a pulse current, unlike the stable output of the graphite/n-Si S-SLG shown in Fig. 1c. The average current $I_{av}$ per cycle and the friction force are depicted in Fig. 1f. The $I_{av}$ of the AFM tip/n-Si ordinary S-G reaches a momentary maximum value of approximately 15 pA, corresponding to a momentary maximum average current density $\sim 4.0 \times 10^4$ Am$^{-2}$ estimated using the Derjaguin–Muller–Toporov (DMT) contact model[37] (see the Methods section and Supplementary Section 5); this is two orders of magnitude larger than that of the S-SLG ($\sim 119$ Am$^{-2}$), but it decays very quickly to $\sim 1$ pA by the 30th cycle. This indicates that

its lifetime is at least two orders of magnitude lower than that of the S-SLG at the 5,000th cycle.

These results are consistent with the high current densities ($10^4 - 10^7$ Am$^{-2}$) and extra short lifetimes (1–5 cycles) reported in the literature[3,21] for all ordinary S-Gs made of nanotip and semiconductor contacts, and reflects the nature of friction-induced excitation. Attempts for high cycle number tests of ordinary S-Gs have been made in terms of macroscale solid–solid contacts, showing stable current densities (0.033–1 Am$^{-2}$) for a large number of cycles of up to 3,600–10,000[15,18]. However, these large cycles correspond to low applied normal pressure values (0.05–5 MPa), low wear, and low friction-induced excitation current[15,18].

**The open-circuit voltage and power measurement of S-SLG.** To further measure the open circuit voltage and power of the S-SLG while accounting for the limitations of the voltage measurement of the AFM system, we built a new measurement system, as shown in Supplementary Fig. 7a. The materials and preparation methods remained unchanged. The top of the graphite flake was pressed using a conductive tungsten probe with a radius of 1 μm. This was controlled by the micromanipulator to induce sliding

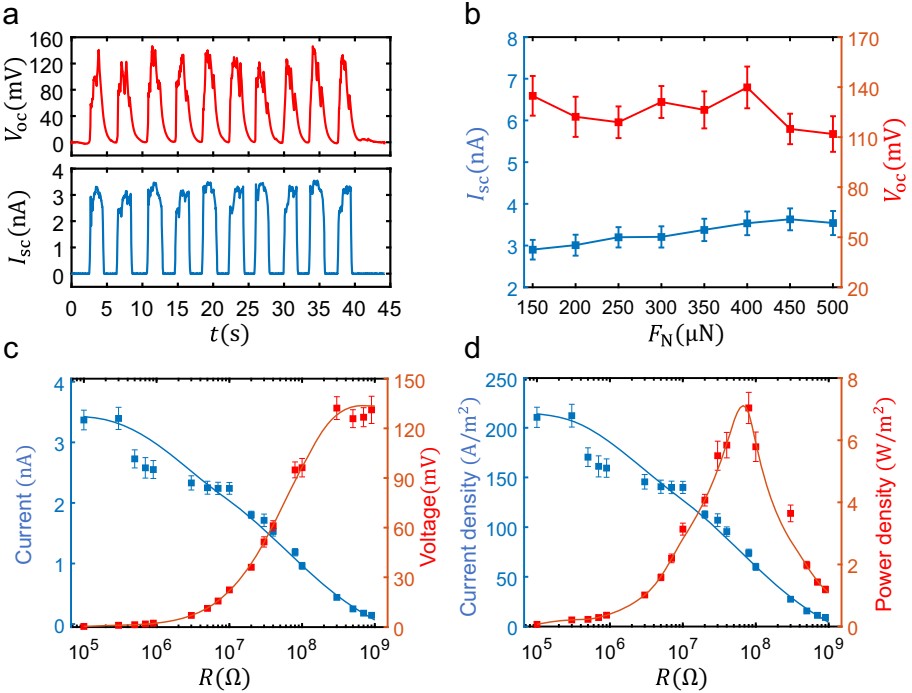

**Fig. 2 Output performance measurement of Schottky superlubric generator (S-SLG) under different normal force and resistance. a** Open-circuit voltage $V_{oc}$ (red) and short-circuit current $I_{sc}$ (blue) waveforms with time $t$ under linear reciprocating motion with speed of 4.3 μm/s and normal force of $F_N =$ 250 μN; the sliding time and pause time are approximately 2 s in each cycle. **b** Relationship between $V_{oc}$ (red) and $I_{sc}$ (blue) with normal force $F_N$ of a graphite/n-Si S-SLG. **c** Output current (blue) and voltage (red) under different resistance $R$. Each data point and error bar in **b** and **c** are obtained by 4000 signal data points. **d** Output current density (blue) and power density (red) under different resistance $R$, each data point and error bar in **d** are calculated by combining the data points in **c** with the contact area $A = 16$ μm².

between the graphite flake and smooth n-Si surface through the piezoelectric displacement control of the micromanipulator. The precision balance at the bottom was used to measure the applied normal force, and an electrometer was connected to the tungsten tip and the Al electrode coated on the bottom of the n-Si to measure the electrical signals (see Methods for more details; the optical microscope image and actual photo of the experimental setup are shown in Supplementary Figs. 7b and c, respectively). The positive direction of the current and voltage measurement was from the graphite flake to n-Si. The open-circuit voltage $V_{oc}$ and short-circuit current $I_{sc}$ waveforms under a linear recipro-cating motion with a speed of 4.3 μm/s and a normal force of $F_N = 250$ μN are shown in Fig. 2a, which is sustainable and stable (the sliding time and pause time are ~2 s in each cycle). Subse-quently, we measured the $V_{oc}$ and $I_{sc}$ waveforms under different $F_N$ as shown in Supplementary Fig. 8, and the average value of $V_{oc}$ and $I_{sc}$ as the function of $F_N$ are shown in Fig. 2b, which has a weak normal force correlation. It is evident from the I–V curves under different normal force in Supplementary Fig. 9 that the additional resistance between graphite flake and n-Si tend to decrease as the normal force increases, resulting in the increase of short-circuit current, but open-circuit voltage will not be affected. Furthermore, we measured the output current and voltage of the S-SLG under different resistances ($R$) while keeping the normal force (250 μN) and speed (4.3 μm/s) constant, as shown in Fig. 2c. When $R < 100$ kΩ, the S-SLG is close to the short-circuit state with a current of 3.4 nA, and when $R > 500$ MΩ, the S-SLG is close to the open-circuit state with voltage of 130 mV. To double check, we further used the null method[20] by changing the bias voltage until the current disappears, to obtain the $V_{oc}$ of the S-SLG, as shown in Supplementary Fig. 10. The obtained

open-circuit voltage $V_{oc}^{(null)} = 138.5$ mV is consistent with the result in Fig. 2 (see Supplementary Section 4.4 for more details), which confirms the accuracy of the measurement. The current and power densities calculated according to Fig. 2c are shown in Fig. 2d with the contact area $A = 16$ μm². The maximum cur-rent density and power density were obtained as 210 Am⁻² and 7 Wm⁻², respectively. Furthermore, the noise current and voltage measurements shown in Supplementary Fig. 7d confirm the reliability of above measurement results.

**Verification of the SSL state at the graphite flake/n-Si interface.** In all reported S-Gs[3,14–18,20–23], evident wear was observed, and the mechanism was mostly explained as friction-induced excitation[20,36,38]. In this mechanism, electrons or holes are excited during sliding owing to the energy released by sliding friction and bonding interaction at the contacted surfaces[23], and be expelled out along the direction of the built-in electric field[19,22,23] or directly tunnel through the Schottky barrier[17,20,39] to result in the DC current.

In our S-SLG, the contact between the graphite flakes and n-Si in the S-SLG is in a state of SSL. To verify this, we conducted a series of tribological tests between a 4 × 4 μm graphite flake with a two-dimensional single crystalline surface (Fig. 3a) and the n-Si surface with atomic smoothness (Fig. 3b); see the Methods section and Supplementary Section 6.1 for details. The measured friction forces under various normal forces are plotted in Fig. 3c and show an extra-low friction coefficient of 0.0039–0.0045. Furthermore, we performed a 6,000-cycle sliding experiment for 4 × 4 μm graphite flakes on n-Si. The measured friction forces of the entire process are shown in Fig. 3d. The friction forces were stabilized in the range of 0.2–0.3 μN after the running-in

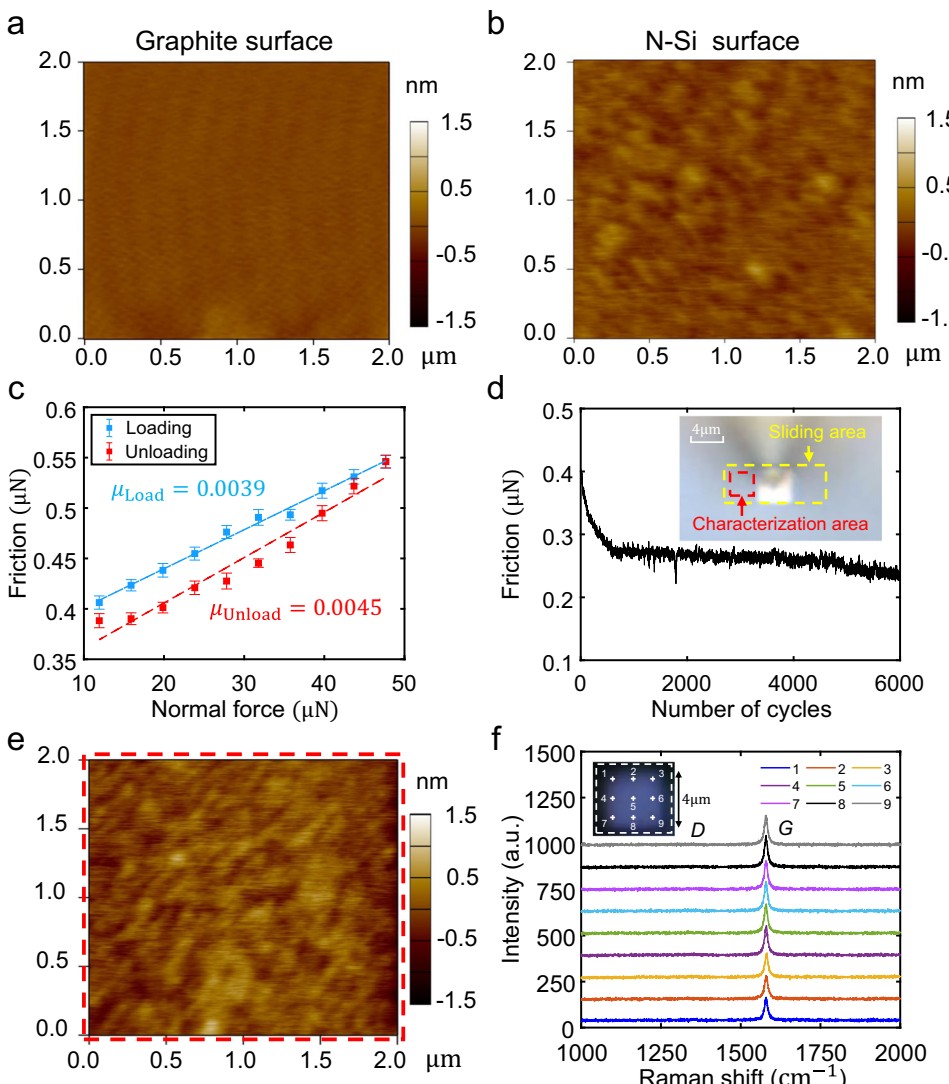

**Fig. 3 Tribological tests between graphite flakes and n-Si. a** Morphological characterization of the two-dimensional single crystal surface of a graphite flake (the colour bar represents height). **b** Morphological characterization of the n-Si surface. **c** Frictional force under different normal forces with a displacement amplitude of 2 μm and speed of 4 μm/s, where blue and red points correspond to loading and unloading processes, respectively; each point and error bar were obtained by 40 tests and had friction coefficients of 0.0039 and 0.0045, respectively. **d** Measured friction force during 6000 continuous sliding cycles with a displacement amplitude of 4 μm and speed of 8 μm/s under a normal force of $N = 23.8$ μN; the illustration is an optical image of the experimental setup, and the yellow dashed frame is the sliding region. **e** Morphological characterization of an n-Si interface after 6,000 sliding cycles; the red dashed frame in **d** is the characterization region. **f** Raman characterization of the graphite flake interface after 6,000 sliding cycles; points 1–9 represent the test positions.

process[35] in the first 600 cycles, which indicates that the SSL state between the graphite flakes and n-Si can be stable for large sliding cycles. After the sliding experiment, we characterized the morphology of the slided n-Si interface (Fig. 3e). There was no obvious wear or damage on the slided n-Si surface. Raman characterization results at different positions (points 1–9 in Fig. 3f) on the slided graphite flake interface and different positions (points 1–6 in Supplementary Fig. 13b) on the slided n-Si interface are displayed in Fig. 3f and Supplementary Fig. 13b, respectively. No observable $D$ peak (1350 cm$^{-1}$) on the slided graphite flake interface and $G$ peak (1580 cm$^{-1}$) on the slided n-Si interface indicates that there was no visible damage to the graphite flake interface after 6,000 sliding cycles (see Supplementary Section 6.2 for the estimation of the graphite defects resolution of Raman measurements). Therefore, the results

confirm the wearless SSL contact state between graphite flakes and n-Si.

**Discussion on the mechanism of S-SLG.** Next, we show that the friction in our S-SLG is insufficient to excite electrons to generate current. According to the friction measurement of graphite/n-Si S-SLG (Fig. 1d), the maximum friction force during the sliding process is approximately $f_{\max}^{(1)} = 0.55$ μN. Friction during superlubric contact mainly occurs along edges through dangling bonds[40]. By approximating the friction force to be evenly distributed along the edge, we estimate the friction force of each dangling bond $f_0^{(1)}$ as $f_0^{(1)} \approx f_{\max}^{(1)} a / 4L = 8.5$ pN, where $a = 0.246$ nm is the lattice constant of the two-dimensional graphite surface. Furthermore, we estimate the upper limit of friction energy $\Delta E_f^{(1)}$ generated by

the interaction of each dangling bond with silicon atoms as $\Delta E_f^{(1)} \approx f_0^{(1)}b = 0.0287\,\text{eV}$, where $b = 0.543\,\text{nm}$ is the lattice constant of silicon. Since $\Delta E_f^{(1)} \ll \Delta E_g = 1.12\,\text{eV}$ and $\Phi_B \approx 0.54\,\text{eV}$, where $\Delta E_g$ and $\Phi_B$ are the band gap of silicon and Schottky barrier height of graphite/n-Si interface, respectively (the $\Phi_B$ is fitted by current–voltage ($I$–$V$) characteristic curve as shown in Supplementary Fig. 9c), it is difficult to excite electrons from valence band to conduction band of n-Si or tunnel from graphite to the conduction band of n-Si through the Schottky barrier (the tunneling probability was calculated as shown in Supplementary Section 7), which indicates that the mechanism of the graphite/n-Si system S-SLG is highly unlikely to be friction-induced excitation.

According to the friction measurement of the AFM tip/n-Si ordinary S-G (Fig. 1f), the minimum friction force during the sliding process is approximately $f_{\min}^{(2)} = 1\,\mu\text{N}$. Using the Hertz's contact theory[41,42], the maximum contact normal stress was calculated as 1.5 times the average contact normal stress. By assuming that the friction shear stress is proportional to the contact normal stress at the contact area $A$ of the tip, we can estimate the maximum friction shear stress as $P_f^{(2)} = 1.5 \times f_{\min}^{(2)}/A \approx 4.37\,\text{GPa}$ (see Supplementary Section 5). Accordingly, the lower limit frictional energy $\Delta E_f^{(2)}$ is $\Delta E_f^{(2)} = P_f^{(2)}c^2b \approx 2.46\,\text{eV}$, where $c \approx 0.408\,\text{nm}$ is the lattice constant of Au. The result of $\Delta E_f^{(2)} > \Delta E_g$

confirms that friction-induced excitation is the main mechanism involved in the AFM tip/n-Si ordinary S-G.

When a metal is in contact with n-Si, electrons are transferred from n-Si to the metal[43]. Consequently, a positively charged region is formed in the n-Si, which is the depletion layer, and a built-in electric field is generated. In the DLED mechanism, sliding of the metal causes the head depletion layer to be established and the tail depletion layer to be destructed, thereby forming built-in electric field separation, which make drifting carriers form a DC current[15,36]. However, to date, the DLED mechanism has not been directly proven experimentally or theoretically. To explain the experimental observations of our S-SLG, we excluded the possibility of friction-induced excitation and instead focused on demonstrating that DLED is the more likely mechanism.

**Quasi-static simulation of DLED mechanism.** As there is still a lack of analysis of the electronic dynamic transportation behaviour of continuous sliding contact in S-SLGs, we performed a quasi-static finite element simulation. As illustrated in Fig. 4a, the model consisted of a slider (length of 4 μm and height of 1 μm) at the top and a much larger stator (length of 20 μm and height of 10 μm) at the bottom, the out-of-plane thickness of model is 1 μm. The stator is made of n-Si with a doping concentration of $N_D = 10^{15}\,\text{cm}^{-3}$. The slider is made of a heavily doped p$^+$-Si as an equivalent metal; the depletion layer penetrates the n-Si, and

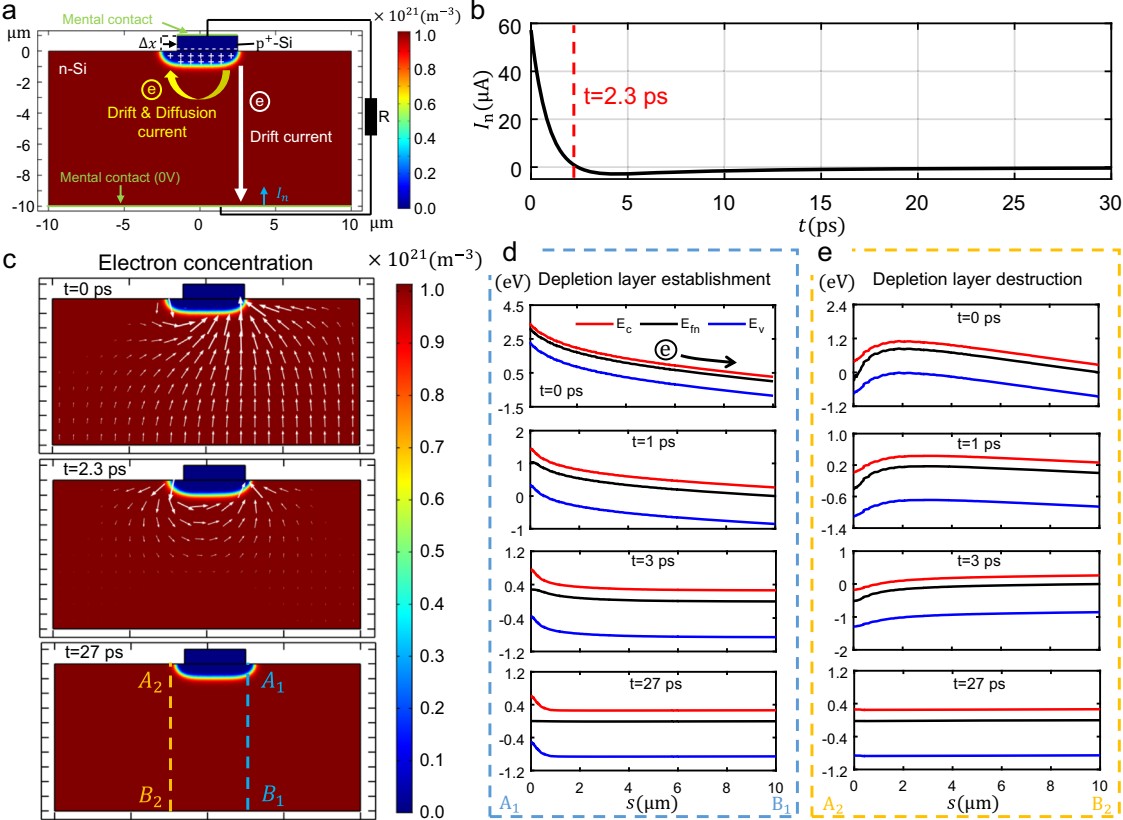

**Fig. 4 Quasi-static semiconductor finite element simulation to explain the physical process of the depletion layer establishment and destruction (DLED) mechanism when $\Delta x = 0.5\,\mu m$. a** Model structure diagram and physical process (the colour bar represent electron concentration). **b** Output electron current $I_n$ along the bottom surface of n-Si with time $t$. **c** Electron concentration distribution with time, where white arrows represent the direction of the electron current (i.e., the reverse direction of the electron motion). **d, e** Distribution of the conduction band energy level $E_c$ (red), valence band energy level $E_v$ (blue), and quasi-electron Fermi level $E_{fn}$ (black) at the $A_1 B_1$ cut line (**d**) ($x = 2.6\,\mu m$, corresponding to the position of the depletion layer establishment) and $A_2 B_2$ cut line (**e**) ($x = -2.1\,\mu m$, corresponding to the position of the depletion layer destruction) in **c**, respectively, where $s$ corresponds to the position.

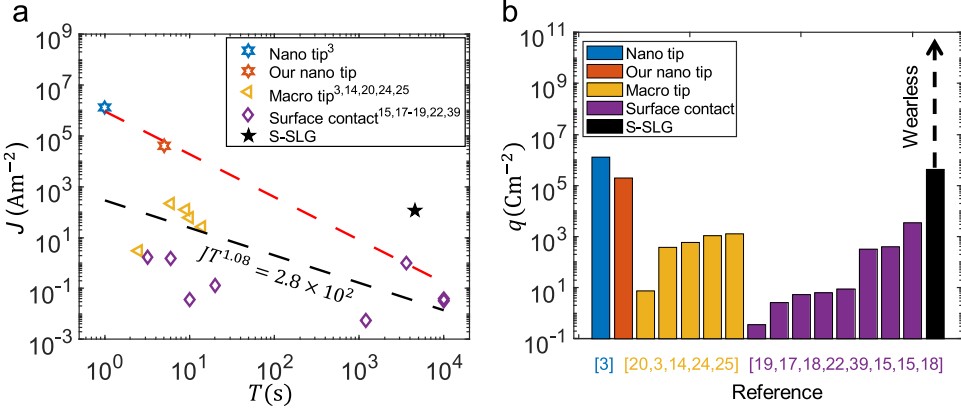

**Fig. 5 Comparison of the Schottky superlubric generator (S-SLG) with reported Schottky generators (S-Gs)[3,14,15,17–20,22,24,25,39]. a** Current density $J$ and lifetime $T$ of S-Gs and S-SLG, where different shapes and colours of dots represent different types of S-G, and black and red dotted lines indicate the power exponential fitting of all points and three upper right points of reported S-Gs, respectively. **b** Transfer charge density $q$, which is the integral of the current density over time ($q = \int_0^T J dt$), of S-Gs and S-SLG, where different colours represent different types of S-G.

the width of depletion in the $p^+$-Si can be neglected, which is similar to metal/semiconductor contact (Supplementary Fig. 15c). The top surface of the slider and the bottom surface of the stator are conductively connected through an external circuit series with a resistance of $R = 10\,\text{k}\Omega$, and the potential of the bottom surface of the stator is set to 0. First, we simulated the electron and hole distribution when the Schottky diode was formed by the slider and the stator (Supplementary Fig. 15b). After equilibrium, we moved the upper slider by a displacement of $\Delta x = 0.5\,\mu\text{m}$ with a contrived constraint that the electron distributions of both the slider and the stator would not change (Fig. 4a). When removing the constraint, the unbalanced electric field drove the electrons to flow along the external circuit and inside the slider and stator and finally to reach a new equilibrium state. Figure 4b shows the output electron current $I_n$ along the bottom surface of the stator with time, and the direction of $I_n$ is shown by the blue arrow in Fig. 4a. A large current occurs in a short period of time ($t < 2.3\,\text{ps}$) when the simulation starts, which is consistent with the direction of the current measured in Fig. 1. The current changes direction at $t = 2.3\,\text{ps}$ and gradually decays to zero. To further explain the cause of this current, Fig. 4c shows the electron concentration distribution at different times (the electric potential distribution is shown in Supplementary Fig. 17a), where the white arrow represents the direction of the electron current; that is, the reverse direction of the electron motion. The conduction band energy level (red), valence band energy level (blue), and electron quasi-Fermi level (black) at $A_1 B_1$ and $A_2 B_2$ in Fig. 4c are shown in Fig. 4d and e, corresponding to the position of the establishment ($x = 2.6\,\mu\text{m}$) and destruction ($x = -2.1\,\mu\text{m}$) of the depletion layer, respectively (the electric fields in the y direction at $A_1 B_1$ and $A_2 B_2$ are shown in Supplementary Figs. 17b and c). When $t = 0\,\text{ps}$, the separation of the built-in electric field causes an unbalanced electric field generated, thereby bending the energy band of entire stator space such as whole $A_1 B_1$ and $A_2 B_2$ cutline (Fig. 4d, e), which causes electronic drift and moves out along the bottom surface of the stator. Therefore, $I_n$ at $t = 0\,\text{ps}$ is contributed by the electronic drift due to the unbalanced electric field. The degree of energy band bending decreases at $t = 1\,\text{ps}$ (Fig. 4d, e), which explains the attenuation of $I_n$ when $0\,\text{ps} < t < 2.3\,\text{ps}$ in Fig. 4b. When $t > 2.3\,\text{ps}$ (i.e., the energy band at $t = 3\,\text{ps}$; Fig. 4d, e), the energy band bending near the bottom surface of the stator disappears and only appears near the depletion layer; that is, the electrons are transferred inside to reach equilibrium rather than the external circuit. Finally, when $t = 27\,\text{ps}$ (Fig. 4d, e), the entire region reaches a new equilibrium state, at which $I_n$

decays to zero, and the energy bands of the $A_1 B_1$ and $A_2 B_2$ cut lines converge to the static equilibrium distribution (quasi-Fermi levels are constant).

The simulation provides a physical image to verify the feasibility and rationality of the DLED mechanism for S-SLG. This physical image indicates that the output current is contributed by the electronic drift caused by the non-equilibrium electric field during the movement of the slider. To further illustrate this point, we obtained the formula $\tau = C^{(\text{eq})}R = SR\sqrt{\frac{eN_D\varepsilon_r\varepsilon_0}{2V_D}} = 3.9\,\text{ps}$ through the theory of Schottky equivalent capacitance[43] (see Supplementary Section 8.4 for more details), where $\varepsilon_r = 11.7$ is the relative permittivity of silicon, $\varepsilon_0$ is the vacuum permittivity, $e$ is the charge of electron, $V_D = 0.87\,\text{V}$ is the contact potential difference and $S = 4\,\mu\text{m}^2$ is the contact area in simulation model. This formula was used to estimate the characteristic time of unbalanced electron drift motion, which is consistent in magnitude with the characteristic time of current decay in Fig. 4b. As the relaxation time of simulation process is much shorter than the characteristic time of the movement, the output current is contributed by the initial drift current ($I_n\,(t = 0)$) when a continuous movement is provided to the slider, as shown in Fig. 4b.

**Comparison between S-SLG and all reported S-Gs.** Next, we compared the S-SLG and all other reported S-Gs to illustrate the advantages of S-SLG. First, we divided all reported S-Gs into nano-tip[3,20], macro tip[3,14,20,24,25], and surface contact[15,17–19,22,39] S-Gs according to their contact area (Supplementary Section 9 and Supplementary Table 1). The current density $J$ and lifetime $T$ of the reported S-Gs are shown in Fig. 5a, and we fitted all the points and the three upper right points (black and red lines, respectively). The inverse proportional relationship ($JT \approx \text{Constant}$) shows that the reported S-G cannot achieve high current density and long lifetime simultaneously. For instance, the nano tip[3,20] S-Gs with the highest current density tend to have the shortest lifetime, as confirmed in our experiments (Fig. 1e, f; red point in Fig. 5a). In contrast, the point obtained by the S-SLG experiment is in the upper right corner of the two lines, which indicates S-SLG can simultaneously achieve high current density and long lifetime for real applications. Furthermore, we integrated the current density over time to obtain the transfer charge density $q = \int_0^T J dt$ of all reported S-Gs and S-SLG (Fig. 5b). S-SLG has a larger transfer charge density than most reported S-Gs, and the achievable

transfer charge density of S-SLG should be larger or unlimited because no current decay or wear was observed during the experiment.

## Discussion

To conclude, we demonstrated the first prototype of a superlubric generator, namely a Schottky superlubric generator (S-SLG), which was made of a microscale graphite flake and n-type silicon (n-Si). This S-SLG not only can stably generate a DC electrical output with high current density of ~210 $Am^{-2}$ and power density of ~7 $Wm^{-2}$, which the current density is three orders of magnitude higher than those of any TENGs (highest reported current density of ~0.1 $Am^{-2}$ [31]) and PENGs (highest current density of ~0.01 $Am^{-2}$ [32]), but more importantly, achieved a long lifetime of at least 5,000 cycles and stable high current density (~119 $Am^{-2}$) simultaneously. Since no wear or current decay in the S-SLG was observed, the S-SLG likely has an unlimited life. Our S-SLG is thus the first microscale generator with both high and stable current density and ultralong or unlimited life. Although Schottky generators based on friction-induced excitation can generate higher current densities ($10^4$–$10^7$ $Am^{-2}$), they have high wear rates and consequently a very short life (a few to dozens of cycles). Our S-SLG has measured the electrical output after excluding the friction-induced excitation mechanism through low friction of the interface and provides a physical image of a conjectured DLED mechanism via quasi-static finite element simulation. The experimental and theoretical results will guide and accelerate real-world applications of S-SLG, such as harvesting vibration energy in human blood vessels to supply power for micro medical robots and high-sensitivity micro-sensors. However, the simulations cannot quantitatively predict the output parameters for continuous sliding processes; therefore, the dynamic process of the DLED mechanism still requires more theoretical research.

## Methods

**Preparation of graphite flake**. We firstly fabricated square graphite mesa arrays with Au film on highly ordered pyrolytic graphite (HOPG, ZYB grade (Brucker)[44]). The fabrication process as shown in Supplementary Fig. 1a, we firstly spin on a double layer photoresist LOR 1 A (100 nm)/ZEP (400 nm) on the fresh cleavage surface of the HOPG (i), and then removed the photoresist of the mesa array area by electron beam lithography (ii). Secondly, we grew the Au array film with thickness of 100 nm through electron beam evaporation (includes the 10 nm Cr as adhesion layer) (iii) and lift-off (iv) process. Lastly, we used metal as a mask to obtain graphite mesa with Au film by reactive ion etching (oxygen ions) process (v), where the etching depth is 2.5 μm. The characterizations of fabricated graphite mesa with Au film are shown in Supplementary Fig. 1b and c. In order to form the S-SLG structure shown in Fig. 1a, we transferred graphite flake with single crystal superlubric interface to the n-Si surface, and the specific process is shown in Supplementary Fig. 2 and Supplementary Section 2.1. We used a tungsten microtip controlled by a micromanipulator (Kleindiek MM3A) as validated by optical microscopy (HiRox KH-3000) to apply a shear stress until they split along their vertical direction, We determine whether the sheared graphite flake undergoes self-recovery motion (SRM)[33] to determine whether it has a single crystal superlubric interface[34].

**Preparation of n-Si**. For the fabrication of n-Si, we uses electron beam evaporation to deposit a layer of 100 nm Al on one side of a 4-inch, 200 um thick, double-polished silicon wafer with a <100> crystal plane, then used the wafer scriber to cut into small pieces of 1 cm × 1 cm size, soaked the each piece in Buffered oxide etch (BOE) solution for 15 min to remove the oxide layer on the surface of n-Si, and finally clean with acetone, alcohol, and deionized water, encapsulate with vacuum.

**S-SLG formation**. We used the microtip to drag the dangling graphite flake controlled by the micromanipulator (Kleindiek MM3A), and placed it slowly by micromanipulator on the atomically smooth fabricated n-Si surface, at this time, since the adsorption force of the graphite flake and n-Si is larger than that of the microtip and graphite flake, the graphite island will remain on the n-Si surface, as shown in Supplementary Fig. 2f, which formed the S-SLG structure shown in Fig. 1a. In order to verify the Schottky contact formed by the graphite flake and n-Si and get the relevant parameters, we performed a static current–voltage (I–V)

characteristic measurement on the transferred S-SLG structure. The detailed experimental and fitting results are shown in Supplementary Fig. 9. The work function of n-Si and HOPG was measured with the AFM tip coated Au (ACCESS-NC-GG(Appnano)) through Asylum Research Cypher S AFM in Scanning Kelvin Probe Microscope (SKPM) model, as details in Supplementary Section 2.2.

**Friction and current measurements of AFM system**. The measurements of the graphite/n-Si heterostructures were performed under an ambient atmosphere. The experimental set-up included a commercial NTEGRA upright AFM (NT-MDT), a 100 μm XYZ piezoelectric displacement platform, a high numerical aperture objective lens (×100 (Mitutoyu)) and visualized conductive AFM tip (ACCESS-NC-GG(Appnano)). Figure 1a shows the schematic of the experimental set-up. We accurately pressed the AFM tip on the Au cap of the graphite flake through the optical microscope and piezoelectric displacement platform. The AFM tip was calibrated in situ by the Sader method[45,46] for the normal direction force and the diamagnetic levitation spring system[47] for the lateral direction force, the specific results of the calibration process are shown in the Supplementary Fig. 4. The bottom Al film of the n-Si is grounded through the iron stage, and the conductive AFM tip is also grounded by connecting a precision ammeter, which can accurately measure the current through AFM tip in the sliding process, and the noise current measurement of NT-MDT AFM system is shown in Supplementary Fig. 5, which is basically maintained at the order of 1 pA.

**Open-circuit voltage and power measurements**. The measurements of the graphite/n-Si heterostructures were performed under an ambient atmosphere. The experimental set-up as shown in Supplementary Fig. 7a included a programmable micromanipulator (Kleindiek MM3A), a precision balance (Mettler) with resolution of 0.1 μN, a high numerical aperture objective lens (−10−×100 (Olympus)) and conductive tungsten probe with radius of 1 μm (prepared by electrochemical corrosion with 5 mol/L KOH solution). Supplementary Figs. 7b and c show the optical microscope image and actual photo of the experimental setup respectively. We accurately pressed the tungsten probe on the Au cap of the graphite flake through the optical microscope and micromanipulator, and accurately measured the applied normal force with the bottom precision balance. The bottom Al film of the n-Si was connected through an electrometer (KEITHLEY 6514) and the tungsten probe, which accurately measured the current and voltage in the sliding process, and the noise current and voltage measurement of above system is shown in Supplementary Fig. 7d, which were basically maintained at the order of 2 pA and 0.5 mV respectively.

**The contact area calculation of AFM tip/n-Si ordinary S-G**. For AFM tip/n-Si ordinary S-G, the DMT model can best approximate the contact between the AFM tip and the hard poorly adhesive material. According to the DMT model[37], the contact area A is given by:

$$A = \pi \left( \frac{R}{K} \left( F_N + 2\pi R\gamma \right) \right)^{\frac{2}{3}}, \quad (1)$$

where R is AFM tip radius, $F_N$ is the normal force apply to the AFM tip, γ is the energy of adhesion, the term $2\pi R\gamma$ can be considered as an additional load, which is determined by the "pull-off" force in the force curve, and $K = \frac{3}{4} \left( \frac{1-\nu_s^2}{E_s} + \frac{1-\nu_t^2}{E_t} \right)$ is the reduced Young's modulus, where $E_t$ and $E_s$ are Young's modulus of tip and sample, and $\nu_t$ and $\nu_s$ are the Poisson ratios of tip and sample, respectively, and the calculation details as shown in Supplementary Section 5. Since the "pull-off" force is much smaller than the normal force applied to AFM tip ($2\pi R\gamma \ll F_N$), we used Hertz contact model[41,42] replace the DMT contact model[37] to consider the normal pressure distribution, as details in Supplementary Section 5, the maximum friction shear stress of the contact region is $P_f = \frac{3}{2} \frac{f}{A}$, where f is the measured friction force.

Surface characterization method of tribological experiment. Here, we show the main process and method of the experiment in Fig. 3. The first step is to use the Asylum Research Cypher S AFM in tapping mode to characterize the topography of the interface (Fig. 3a) of selected graphite flake with SRM[33] property through flipped 180 degree of microtip after adsorbing the graphite flake (Supplementary Fig. 2e) before placing it on the n-Si surface (Supplementary Fig. 2f), and characterized the topography of n-Si surface (Fig. 3b) at the same time. For the second step, we used the lateral force measurement system of AFM (NT-MDT) to measure the coefficient of friction (Fig. 3c), and then measured the friction force of 6,000 cycles sliding process (Fig. 3d). For the third step, we firstly used Asylum Research Cypher S AFM in tapping mode to perform a larger range morphological characterization and find the position of the graphite flake as shown in Supplementary 12, and further characterize the small sliding region of n-Si through the positioning function of AFM, to judge whether there is any observable damage (Fig. 3e). For the fourth step, we performed a Raman characterization (LabRAM HR Evolution Raman spectrometer from HORIBA, with resolution of 0.1 $cm^{-1}$, laser wavelength of 532 nm, grating of 1800 (450–850 nm), acq. time of 4 s and spot diameter of 1 μm) on the slided n-Si interface (Supplementary Fig. 13), to judge whether there is any observable graphite wear debris from whether there is a G peak (1580 $cm^{-1}$). Lastly, we used the method shown in Supplementary Fig. 14 to lift the graphite flake by overcoming the van der Waals adsorption force between graphite flake and n-Si interfaces, and flipped 180 degree to perform the Raman characterization (Horiba)

on the slided graphite flake interface (Fig. 3f), to judge whether there is any observable damage from whether there is a $D$ peak ($1350\,cm^{-1}$).

**The setup of quasi-static simulation of DLED mechanism.** The geometric parameters and boundary conditions of the model were set as shown in Supplementary Fig. 15a, the model consists of a slider (length 4 μm and height 1 μm) at the top and a much larger stator (length 20 μm and height 10 μm) at the bottom, the out-of-plane thickness of model is 1 μm. The stator is made of n-Si with a doping concentration of $N_D = 10^{15}\,cm^{-3}$. The slider is made of a heavily doped p$^+$-Si with a doping concentration of $N_A = 2 \times 10^{19}\,cm^{-3}$ as an equivalent metal. The interface between the stator and the slider (yellow solid line) was set to a continuous heterojunction condition

$$
\begin{aligned}
E_{fn}^{(1)} &= E_{fn}^{(2)}, \\
E_{fp}^{(1)} &= E_{fp}^{(2)}, \\
\overrightarrow{D_1} &= \overrightarrow{D_2},
\end{aligned}
\tag{2}
$$

where $E_{fn}^{(i)}$ and $E_{fp}^{(i)}$ ($i = 1, 2$ represents the slider and stator) are the fermi levels of electrons and holes respectively. The top surface of the slider and the bottom surface of slider were set to metal contact boundary, which are conductively connected through an external circuit series with a resistance $R$, the potential of the bottom surface of stator was set to be 0, and the potential of the top surface of the slider $V_1$ was determined according to the continuous conditions of the external current $I_R$ through resistance

$$
\begin{aligned}
V_1 &= I_R R, \\
I_R &= \iint I_n dS,
\end{aligned}
\tag{3}
$$

where $I_n$ is the normal current on the bottom surface of the stator. The potential distribution $V$ in the semiconductor satisfies the Poisson equation

$$
\nabla \cdot (\varepsilon \nabla V) = q(n - p - N_D + N_A),
\tag{4}
$$

where $\varepsilon$ is the permittivity, $n$ and $p$ are the electron and hole concentration respectively. The relationship between carrier concentration and energy band is given by statistical theory

$$
\begin{aligned}
n &= N_c \exp\left(-\frac{E_c - E_{fn}}{kT}\right), \\
p &= N_v \exp\left(\frac{E_v - E_{fp}}{k_B T}\right), \\
E_c &= -\chi_{n-Si} - qV, \\
E_v &= -\chi_{n-Si} - E_g - qV,
\end{aligned}
\tag{5}
$$

where $N_c = 2\left(\frac{2\pi m_e^* k_B T}{h^2}\right)^{\frac{3}{2}}$ and $N_v = 2\left(\frac{2\pi m_p^* k_B T}{h^2}\right)^{\frac{3}{2}}$ are the thermally excited state density of electron and hole, $E_c$ and $E_v$ are the conduction band and valence band energy levels, $E_g = 1.12$ eV and $\chi_{n-Si} = 4.05$ eV are the band gap and electron affinity of n-Si respectively. By solving equations Eq. (4) and Eq. (5), we obtained the potential distribution and carrier concentration distribution in the semiconductor, and further, we calculated the electron and hole currents ($\overrightarrow{J_n}$ and $\overrightarrow{J_p}$) by the drift diffusion model

$$
\begin{aligned}
\overrightarrow{J_n} &= -qn\mu_n \nabla V + qD_n \nabla n, \\
\overrightarrow{J_p} &= -qn\mu_p \nabla V - qD_p \nabla p, \\
\nabla \cdot \overrightarrow{J_n} &= q\frac{\partial n}{\partial t}, \\
\nabla \cdot \overrightarrow{J_p} &= -q\frac{\partial p}{\partial t},
\end{aligned}
\tag{6}
$$

where $\mu_n$ and $D_n$ are the electron mobility and diffusion coefficient respectively, the $\mu_p$ and $D_p$ are the hole mobility and diffusion coefficient respectively.

**The simulation process.** The whole quasi-static simulation process was divided into three steps: Step 1: we simulated the electron and hole distribution when the Schottky diode was formed to reach static equilibrium ($\Delta x = 0$) by the slider and stator, as shown in Supplementary Fig. 15b. Step 2: After equilibrium, we moved the upper slider with a displacement of $\Delta x = 0.5\,\mu m$ with a contrived constraint that the electron distributions of both the slider and the stator would not change, as shown in Fig. 4a, and the carrier distribution had reached a non-equilibrium state. Step 3: We removed the constraint and start the transient simulation with the state of the step 2 as the initial condition, to simulate the transport process of carriers, electric potential distribution and related parameters (energy band distribution of cutline $A_1 B_1$ and $A_2 B_2$) over time.

## Data availability

The data that support the findings of this study are available from the corresponding author upon reasonable request. Source data for Figs. 1c–f, 2, 3, 4 and Supplementary Figs. 3, 4, 5, 6e, 7d, 8, 9, 10b–d, 11, 13, 15b, c, 17 are provided with the paper. Source data are provided with this paper.

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

## Acknowledgements

We wish to acknowledge the financial support of the National Natural Science Foundation of China (Grant Nos. 11572173, 11890671, 51961145304), National Key Basic Research Program of China (Grant No. 2013CB934200), Cyrus Tang Foundation (Grant No. 202003), Beijing Municipal Science & Technology Commission (Program No. Z151100003315008), Tsinghua University Initiative Scientific Research Program (Grant Nos. 2014z01007 and 2012Z01015), and State Key Laboratory of Tribology Tsinghua University Initiative Scientific Research Programs (Grant No.SKLT2019D02). We would also like to acknowledge Shisheng Lin from Zhejiang University and Zhonglin Wang from Beijing Institute of Nanoenergy and Nanosystems for their valuable comments. We also thank Chi Zhang and Zhi Zhang from Beijing Institute of Nanoenergy and Nanosystems for their suggestions and supports. Finally, we would like to acknowledge Editage (www.editage.cn) and Tian Gan from Tsinghua University for English language editing.

## Author contributions

Q.Z. and X.H. designed the experimental aspects of the study, X.H. and X.J. performed the experiments and analysed the experimental data; J.N. assisted in electrical output measurement under different resistance, and D.P. assisted in tribological experiments and characterization; X.H. and F.Y. designed and conducted the quasi-static simulations of DLED mechanism; Z.W. conducted the simulation of pressure distribution of graphite flake; X.H., H.J., and X.P. analysed the simulation results and mechanism. All authors contributed to the writing of this manuscript.

## Competing interests

The authors declare no competing interests.
