## [Peer Review File · Nature Communications]

REVIEWER COMMENTS

Reviewer #1 (Remarks to the Author):

This work proposes a microscale Schottky superlubric generator (S-SLG) with sliding contact between micro-sized graphite flakes and n-type silicon. The S-SLG generates a stable electrical current at high density ($\sim 119 \text{ Am}^{-2}$) for at least 5,000 cycles, which is potential in real-world applications. Especially, the authors exclude the friction excitation mechanism of current generation, and reveal the mechanism of depletion layer establishment and destruction (DLED) through finite element simulations. This work not only accelerates the potential use of S-SLGs in real-world applications, but also guides its work mechanism based on semiconductor physical. Considering its high quality and fitting with the scope of "Nature communications", publication after a major revision is recommended. Below are my questions or suggestions which may be helpful for revision:

1. The reference 15 seems have no relation with sliding Schottky junctions based direct current generator?
2. As the authors mentioned in 2nd paragraph of the manuscript, "corresponding to low friction (with respect to a normal pressure of 2–5 kPa) and current densities of 0.0055–0.13 Am^{-2} , which are too low for most applications^{14,19-21}". But as shown in the ref. [14, 19-21], the current density is $\sim 100 \text{ Am}^{-2}$, which should be adjusted.
3. As shown in Figure 2, the N-Si surface seems have no obvious abrasion during the sliding cycles. How about the layered graphite flakes? As a typical layered semiconductor, will the lifetime of generator be longer when the MoS₂ be used to replace the N-Si substrate? The author can explore it in the further work.
4. As shown in Figure 3, an electronic drift process during relative sliding was explored using a quasi-static semiconductor finite element simulation. So the response time of the current generating process is as fast as $\sim 3 \text{ ps}$? And the arrows in Figure c are not obvious enough.
5. As a generator, the power density and internal resistance are always measured. The author can measure it with different load resistance.

Reviewer #2 (Remarks to the Author):

The authors reported a Schottky DC generator with high performance and long-term durability based on superlubric sliding system. It is a significant contribution in terms of bridging the gap between size-effect limited DC generation and the scaled-up power generation for practical applications. Moreover, the authors provided a unique angle in explaining the DC generation phenomenon in the Schottky sliding contacts, which would be of great interest for both materials scientists and physicists. Therefore, I would suggest the accepting of the manuscript with proper addressing of the following questions and comments:

Questions:

1. From an intuitive perspective, it seems like the surface charged region (SCR, either depletion/accumulation/inversion) are building up and breaking down at the same time. This is supposed to generate zero net current. Could the authors explain the driving force and the electron source for the net current generation?
2. The mechanism analysis using dangling bond force and energy for each surface atoms is reasonable. The authors estimated that the energy per bond to be $\sim 0.0287 \text{ eV}$, which correspond to $\sim 28 \text{ mV}$ per electron, which I would also agree with.

However, note that the energy barrier (Schottky barrier, on the order of 100 meV) for charge transport across the MS junction is much lower than the Si band gap value (1.1 eV), which could

be further lowered by image charge effect and surface state transport mechanism. Therefore, I am concerned about the complete exclusion of excitation mechanism from the picture (may be a coupling effect of both mechanisms with different weight in tip/plane, and 2D/plane superlubric system?)

3. Have the authors measured the open-circuit voltage by null method (Nano Energy 48, 320-326)? This might shed some light regarding point 2.

4. Could the authors envision the possible scenario for future applications? (For example, harvesting energy from bearing movement with 2D lubricant?)

Comments:

1. It is suggested that the 'Direct-current' to be included in the title to emphasize the key feature of the generator (i.e. Microscale Superlubric Schottky Direct-current Generator with High Current Density and Ultralong Life)

2. Some other relevant literatures (Schottky/piezoelectric/flexoelectric generator based on 2D materials) may be added and worth being discussed:

1) Liu, et al. 'Scaled-up Direct-Current Generation in MoS₂ Multilayer-Based Moving Heterojunctions', ACS applied materials & interfaces 2019 11 (38), 35404-35409;

2) Cai, H, et al. 'Tribo-Piezoelectricity in Janus Transition Metal Dichalcogenide Bilayers: A First-Principles Study'. Nano Energy 2019, 56, 33–39.

3) Zhu, et al. 'Observation of piezoelectricity in free-standing monolayer MoS₂', Nature Nanotechnology volume 10, pages151–155(2015)

Reviewer #3 (Remarks to the Author):

The study presents a combined experimental and theoretical exploration of the electricity generating in sliding graphene/n-type silicon system. Design of experiment involving transfer of graphitic mesas on silicon substrate and their lateral movement leads to establishment of a superlubricity regime that consequently results in a stable current density outcome. The authors correlate the observations with the depletion layer establishment and destruction (DLED) during sliding. Theoretical analysis further supports the experimental observations.

I have a two-fold feeling from the paper. Overall it is very nicely written, provides very clear analysis of the observations and their discussion, and concludes with a proposed fundamental mechanism that fits well. From another side, after looking at the previous literature, some statements are exaggerated and the proposed design largely replicates previous studies. There is still a room for improvement to clearly highlight the novelty of the proposed study without lessening previous contributions.

For example, the authors state that the observed in their work current density is several orders of magnitude higher than any previous work. However, some earlier papers reported very similar S-G concept of using graphene on silicon and even the current density was of the similar value (Look for example at Lin, Shisheng, et al. Advanced Materials 31.7 (2019): 1804398; the reported current density was 40 Am⁻²). As a result, the novelty of this study is a not very clear.

The statements in the Abstract and Intro on importance of micro electrical generators for IOT, Networking, Data analysis, etc are too ambitious and general and thus sound irrelevant for the paper concept.

Some experimental procedure steps lack clarification: what photoresist was used, how the contamination from the photoresist removal may affect the quality of the graphite flakes? How much effect comes from the load during sliding?

It is very intriguing to see that there is a superlubricity state between graphite and silicon with no signs of wear considering high surface energy of the cleaned silicon substrate and defects with dangling bonds on the edges of graphene flakes. An additional discussion of the observed effect would be helpful. If the transfer of graphene film onto the silicon substrate eventually occurs how would it affect the electricity generation? Would it lead to the failure of the system?

Figure 1d indicates that there is a correlation between the friction and the current, thus suggesting that it is not purely DLED effect but rather a synergy of two effects. This might be better observed from studies involving changing the applied load.

Recheck for typos both in the manuscript and in the supplementary info.

Below we respond (in Black) all points (in Italic) raised by the referees:

Reviewer #1

*Comment: “This work proposes a microscale Schottky superlubric generator (S-SLG) with sliding contact between micro-sized graphite flakes and n-type silicon. The S-SLG generates a stable electrical current at high density ($\sim 119 \text{ Am}^{-2}$) for at least 5,000 cycles, **which is potential in real-world applications**. Especially, the authors exclude the friction excitation mechanism of current generation, and reveal the mechanism of depletion layer establishment and destruction (DLED) through finite element simulations. **This work not only accelerates the potential use of S-SLGs in real-world applications, but also guides its work mechanism based on semiconductor physical. Considering its high quality and fitting with the scope of "Nature communications", publication after a major revision is recommended.** Below are my questions or suggestions which may be helpful for revision:”*

Response: We would like to thank Reviewer 1 for his/her careful reading and positive comments.

1. The reference 15 seems have no relation with sliding Schottky junctions based direct current generator?”

Response: Note that we have deleted this reference in the revised manuscript.

“2. As the authors mentioned in 2nd paragraph of the manuscript, “corresponding to low friction (with respect to a normal pressure of 2–5 kPa) and current densities of $0.0055\text{--}0.13 \text{ Am}^{-2}$, which are too low for most applications^{14,19-21}”. But as shown in the ref. [14, 19-21], the current density is $\sim 100 \text{ Am}^{-2}$, which should be adjusted.”

Response: Herein, we wanted to point out that in all previous experiments, to the

best of our knowledge, the long lifetimes and high current densities had not been simultaneously achieved. For instance, the longest lifetimes of S-Gs (in the range of 3,600–10,000 cycles^{1,2} realized with respect to a low normal pressure of 0.05–5 MPa) can only generate fairly low current densities (approximately 0.033–1 Am⁻²). In contrast, the highest current density $\sim 40 - 112 \text{ Am}^{-2}$ in the reference^{1,2} was not measured in a state with a long lifetime. The above clarification has been added in Introduction part in revised manuscript.

“3. As shown in Figure 2, the N-Si surface seems have no obvious abrasion during the sliding cycles. How about the layered graphite flakes? As a typical layered semiconductor, will the lifetime of generator be longer when the MoS₂ be used to replace the N-Si substrate? The author can explore it in the further work.”

Response: We obtained the Raman characterization results at different positions (points 1–9 in Fig. 3f) on the slided graphite flake interface and different positions (points 1-6 in Supplementary Fig. S14b) on the slided n-Si interface, as shown in Fig. 3f and Supplementary Fig. S14b, respectively. Note that we did not observe any D peak (1350 cm⁻¹) on the slided graphite flake interface and G peak (1580 cm⁻¹) on the slided n-Si interface, suggesting that there was no visible damage on graphite flake interface after 6,000 sliding cycles. The detailed discussion of the above results has been added in subsection “The verification of SSL state at graphite flake/n-Si interface” of the revised manuscript.

As you suggested, the proposed study shows that graphite flake and MoS₂ with the single crystal two-dimensional surface can achieve a robust structural superlubricity³. We believe that S-SLG under the abovementioned structure can achieve long lifetime. We will carefully explore it in the further work.

“4. As shown in Figure 3, an electronic drift process during relative sliding was explored using a quasi-static semiconductor finite element simulation. So the response time of the current generating process is as fast as $\sim 3 \text{ ps}$? And the arrows in Figure c

are not obvious enough.”

Response: To understand this characteristic time ~ 3 ps and its corresponding physical image, we obtained the formula $\tau = C^{(eq)}R = SR\sqrt{\frac{eN_D\epsilon_r\epsilon_0}{2V_D}} = 3.9$ ps through the theory of Schottky equivalent capacitance⁴ (see Supplementary Section 8.4 for more details), where $\epsilon_r = 11.7$ is the relative permittivity of silicon, ϵ_0 is the vacuum permittivity, e is the charge of electron, $V_D = 0.87$ V is the contact potential difference and $S = 4$ μm^2 is the contact area in simulation model. This formula was used to estimate the characteristic time of unbalanced electron drift motion, which is consistent in magnitude with the characteristic time of current decay in Fig. 4b. We have added this discussion in subsection “Quasi-static simulation of DLED mechanism” of the revised manuscript.

The new Fig. 4b has been indicated by a bold arrow in the revised manuscript.

“5. As a generator, the power density and internal resistance are always measured. The author can measure it with different load resistance.”

Response: We built a new experimental set-up that measured the output current, voltage, and power of our Schottky superlubric generator (S-SLG) under different resistances; these results have been added as Fig. 2 in the revised manuscript.

Reviewer #2

Comment: “The authors reported a Schottky DC generator with high performance and long-term durability based on superlubric sliding system. It is a significant contribution in terms of bridging the gap between size-effect limited DC generation and the scaled-up power generation for practical applications. Moreover, the authors provided a unique angle in explaining the DC generation phenomenon in the Schottky sliding

contacts, which would be of great interest for both materials scientists and physicists. Therefore, I would suggest the accepting of the manuscript with proper addressing of the following questions and comments.”

Response: We would like to thank Reviewer 2 for his/her careful reading and positive comments.

“Questions:

1. From an intuitive perspective, it seems like the surface charged region (SCR, either depletion/accumulation/inversion) are building up and breaking down at the same time. This is supposed to generate zero net current. Could the authors explain the driving force and the electron source for the net current generation?”

Response: Considering the DLED mechanism, we believe that the formation and destruction of the depletion layer occur simultaneously, which can be seen from the dynamic process of the electron concentration distribution and direction of the electron flow in Fig. 4c. However, in the above process, the separation of built-in electric field results in an unbalanced electric field generated (the electric potential distribution is shown in Supplementary Fig. S18a), thereby bending the energy band of entire stator space, such as the entire A_1B_1 and A_2B_2 cutline (Fig. 4d,4e); the electric fields in the y direction at A_1B_1 and A_2B_2 are shown in Supplementary Figs. S18b and c. This unbalanced electric field caused by relative sliding will result in electronic drift and will move out along the bottom surface of the stator to generate current.

“2. The mechanism analysis using dangling bond force and energy for each surface atoms is reasonable. The authors estimated that the energy per bond to be ~ 0.0287 eV, which correspond to ~ 28 mV per electron, which I would also agree with. However, note that the energy barrier (Schottky barrier, on the order of 100 meV) for charge transport across the MS junction is much lower than the Si band gap value (1.1 eV),

which could be further lowered by image charge effect and surface state transport mechanism. Therefore, I am concerned about the complete exclusion of excitation mechanism from the picture (may be a coupling effect of both mechanisms with different weight in tip/plane, and 2D/plane superlubric system?)”

Response: Please note that to address this question, we have added some relevant analyses. First, we measured the current–voltage (I–V) characteristic curve of S-SLG, as shown in Supplementary Fig. S3, where the Schottky barrier height $\Phi_B = 0.67$ eV is fitted by thermionic emission model, as detailed in Supplementary Section 2.1. Second, we calculated the tunnelling probability of electron through Schottky barrier based on the principle of tunnelling effect, as shown in Supplementary Section 7. Considering the upper limit of friction energy obtained by electrons is $\Delta E_f^{(1)} = 0.0287$ eV, the tunnelling probability is calculated as $T_t \approx 10^{-546}$ (as detailed in Supplementary Section 7), which indicates the mechanism of friction-induced tunneling current is virtually impossible to occur in the proposed S-SLG. This discussion has been added in subsection “Discussion on the mechanism of S-SLG” in the revised manuscript.

“3. Have the authors measured the open-circuit voltage by null method (Nano Energy 48, 320-326)? This might shed some light regarding point 2.”

Response: We built a new experimental set-up that directly measures the open-circuit voltage of our S-SLG, as added in Fig. 2 in the revised manuscript. For confirming this, we used the null method⁵ (Nano Energy 48, 320-326) by changing the bias voltage until the current disappears to obtain the V_{oc} of S-SLG, as shown in Supplementary Fig. S11. The obtained open-circuit voltage $V_{oc}^{(null)} = 138.5$ mV is consistent with the result in Fig. 2 (see Supplementary Section 4.4 for more details), which confirms the accuracy of the measurement. These results have been added in subsection “The open-circuit voltage and power measurement

of S-SLG” in the revised manuscript.

“4. Could the authors envision the possible scenario for future applications? (For example, harvesting energy from bearing movement with 2D lubricant?)”

Response: We have added some examples of potential applications in the Conclusion section of the revised manuscript, such as harvesting vibration energy in human blood vessels to supply power for micro medical robots and high-sensitivity micro-sensors. Considering that this study primarily aims to propose such a brand-new microscale generator structure, the possibility of application will be carefully studied in further research.

“Comments:

1. It is suggested that the ‘Direct-current’ to be included in the title to emphasize the key feature of the generator (i.e. Microscale Superlubric Schottky Direct-current Generator with High Current Density and Ultralong Life)”

Response: As per your suggestion, we have revised the title of the manuscript to “Microscale Schottky Superlubric Generator with High Direct-current Density and Ultralong Life.”

“2. Some other relevant literatures (Schottky/piezoelectric/flexoelectric generator based on 2D materials) may be added and worth being discussed:

1) Liu, et al. ‘Scaled-up Direct-Current Generation in MoS₂ Multilayer-Based Moving Heterojunctions’, ACS applied materials & interfaces 2019 11 (38), 35404-35409;

2) Cai, H, et al. ‘Tribo-Piezoelectricity in Janus Transition Metal Dichalcogenide Bilayers: A First-Principles Study’. Nano Energy 2019, 56, 33–39.

3) Zhu, et al. ‘Observation of piezoelectricity in free-standing monolayer MoS₂’, Nature Nanotechnology volume 10, pages151–155(2015)”

Response: The relevant literatures about Schottky generator based on 2D materials have been discussed and added in the revised manuscript.

Reviewer #3

*Comment: “The study presents a combined experimental and theoretical exploration of the electricity generating in sliding graphene/n-type silicon system. Design of experiment involving transfer of graphitic mesas on silicon substrate and their lateral movement leads to establishment of a superlubricity regime that consequently results in **a stable current density outcome**. The authors correlate the observations with the depletion layer establishment and destruction (DLED) during sliding. **Theoretical analysis further supports the experimental observations.**”*

Response: We would like to thank Reviewer 3 for his/her careful reading and positive comments.

*“I have a two-fold feeling from the paper. Overall it is very nicely written, provides very clear analysis of the observations and their discussion, and concludes with a proposed fundamental mechanism that fits well. From another side, after looking at the previous literature, some statements are exaggerated and **the proposed design largely replicates previous studies**. There is still a room for improvement to clearly highlight the novelty of the proposed study **without lessening previous contributions.**”*

*“For example, the authors state that the observed in their work current density is several orders of magnitude higher than any previous work. However, some earlier papers reported very similar S-G concept of using graphene on silicon and even the current density was of the similar value (Look for example at Lin, Shisheng, et al. *Advanced Materials* 31.7 (2019): 1804398; the reported current density was 40 Am⁻²). As a result, the novelty of this study is a not very clear.”*

Response: We would like to thank the reviewer for this constructive suggestion. We made improvement to clearly highlight the novelty in mainly the following two aspects:

First, to the best of our knowledge, this is the first study to propose Schottky superlubric generators (S-SLG). The S-SLG simultaneously achieved a long lifetime of ~5,000 cycles and high current density of ~119 Am⁻². In comparison, in all the previously conducted studies, to the best of our knowledge, the long lifetimes and high current densities had not been simultaneously achieved. For instance, the longest lifetimes of S-Gs (in the range of 3,600–10,000 cycles^{1,2} realized with respect to a low normal pressure of 0.05–5 MPa) can only generate a fairly low current densities (approximately 0.033–1 Am⁻²). In contrast, the highest current density ~40 – 112 Am⁻² in the reference^{1,2} was not measured in a state with a long lifetime (typically several cycles).

Second, there were two mechanisms of generating electricity of Schottky generators, namely the friction-induced excitation and demonstrate an electronic drift process (DLED). Using S-SLG, for the first time, we showed that without the friction-induced excitation, the S-SLG can still generate electricity. We further demonstrated an electronic drift process (DLED) during relative sliding using a quasi-static semiconductor finite element simulation.

“The statements in the Abstract and Intro on importance of micro electrical generators for IOT, Networking, Data analysis, etc are too ambitious and general and thus sound irrelevant for the paper concept.”

Response: We have revised the Abstract and Introduction sections to enhance the novelty description. For instance, we modified the corresponding part of the abstract to “Miniaturized or microscale generators that can effectively and persistently convert weak and random mechanical energy into electricity have significant potential to provide solutions for the power supply problem of distributed sensors and devices.”

“Some experimental procedure steps lack clarification: what photoresist was used, how the contamination from the photoresist removal may affect the quality of the graphite flakes? How much effect comes from the load during sliding?”

Response: We have added the photoresist model information in the Method section of the revised manuscript.

The graphite surface in superlubric contact with the n-Si film was cleaved after the photoresist removal process. Previous studies⁶⁻⁹ proved that all such cleavage surfaces with the superlubricity were absolutely clean, without any contaminations.

In the experiment shown in Fig. 1, we did not change the applied normal force during the sliding process. We built a new experimental set-up that measured the open-circuit voltage and short-circuit current under different normal force, as shown in Fig. 2b in the revised manuscript. The results indicate the weak correlation between the load and current.

“It is very intriguing to see that there is a superlubricity state between graphite and silicon with no signs of wear considering high surface energy of the cleaned silicon substrate and defects with dangling bonds on the edges of graphene flakes. An additional discussion of the observed effect would be helpful. If the transfer of graphene film onto the silicon substrate eventually occurs how would it affect the electricity generation? Would it lead to the failure of the system?”

Response: Some previously reported studies have investigated the superlubricity in nanoscale contacts between a graphene and atomic smooth three-dimensional material such as diamond-like-carbon (DLC)¹⁰ and gold¹¹. Recently, our

experiments showed that the superlubricity occurred between a microscale graphite fully contacted with a variety of atomically flat three-dimensional materials (such as sapphire, mica, DLC, silicon, HfO₂, Al₂O₃, and SiO₂), with coefficients of friction as low as 0.003–0.006, which are in the superlubricity range. We further found the wear-free sliding for a micrometer-sized graphite flake on a DLC surface under ambient condition with speeds up to 2.5 m/s and distance over 100 km. The study that shows these latest results and their superlubricity mechanism is under submission/review process¹². Detailed explanation is out of the scope of the current paper.

If the transfer of graphene film onto the silicon substrate occurs eventually, we think that the output will decay or even disappear if there is only relative sliding between graphite flake and transferred graphene. In fact, we have experimentally observed the decay and disappearance of the current after the graphite flake layered, as shown in Supplementary Fig. S7, thereby confirming that the current is caused by the sliding of graphite/n-Si interface instead of graphite/graphite interface. This also illustrates from the side that the long lifetime and high-density current output of the S-SLG, as shown in Fig. 1, did not occur the transfer of graphene film to silicon.

“Figure 1d indicates that there is a correlation between the friction and the current, thus suggesting that it is not purely DLED effect but rather a synergy of two effects. This might be better observed from studies involving changing the applied load.”

Response: According to the friction measurement of graphite/n-Si S-SLG in Fig. 1d, the maximum friction force during the sliding process is approximately only 0.55 μN. By approximating the friction force to be evenly distributed along the edge, we estimated the upper limit of friction energy generated by the interaction of each dangling bond with silicon atoms as 0.0287 eV, which is much smaller than the band gap of silicon ($\Delta E_g = 1.12$ eV) and Schottky barrier height of

graphite/n-Si interface ($\Phi_B = 0.67$ eV), indicating that the mechanism of the graphite/n-Si system S-SLG is highly unlikely to be friction-induced excitation. The detailed analyses are shown in “Discussion on the mechanism of S-SLG” section in the revised manuscript. Therefore, the current increase, as shown in Fig. 1d, should be completely based on the DLED mechanism, rather than friction-induced mechanism, and the slight increase in friction can be attributed to the increase in speed.

To estimate the influence of normal force, we added measurements of open-circuit voltage and short-circuit current under different normal force, as shown in Fig. 2e in the revised manuscript. The linear fit slope of the short-circuit current with respect to the applied normal force is $s^{(N)} = 2.08 \times 10^{-3}$ nA/uN, which shows a weak normal force correlation. As a by-proof, we also measured the I-V curves and fitted the ideal factor in Supplementary Fig. S10. The results show that the contact between graphite flake and n-Si tend to the ideal Schottky junction (ideal factor decrease) with an increase in the normal force^{13,14}, thus indicating a better drift output current under DLED mechanism¹³.

“Recheck for typos both in the manuscript and in the supplementary info.”

Response: We have carefully rechecked both the revised manuscript and supplementary information for any typos.

References

- 1 Lin, S., Lu, Y., Feng, S., Hao, Z. & Yan, Y. A High Current Density Direct-Current Generator Based on a Moving van der Waals Schottky Diode. *Advanced Materials* **31**, doi:10.1002/adma.201804398 (2019).
- 2 Lu, Y. *et al.* Direct-Current Generator Based on Dynamic PN Junctions with the Designed Voltage Output. *Iscience* **22**, 58-69, doi:10.1016/j.isci.2019.11.004 (2019).
- 3 Song, Y. *et al.* Velocity dependence of friction in single crystalline superlubric heterojunctions: The case of graphite/molybdenum disulfide interface.(In

submission)

- 4 Zhang, Z. & Yates, J. T., Jr. Band Bending in Semiconductors: Chemical and Physical Consequences at Surfaces and Interfaces. *Chemical Reviews* **112**, 5520-5551, doi:10.1021/cr3000626 (2012).
- 5 Liu, J. *et al.* Sustained electron tunneling at unbiased metal-insulator-semiconductor triboelectric contacts. *Nano Energy* **48**, 320-326, doi:10.1016/j.nanoen.2018.03.068 (2018).
- 6 Liu, Z. *et al.* Observation of Microscale Superlubricity in Graphite. *Physical Review Letters* **108**, 065502, doi:10.1103/PhysRevLett.108.205503 (2012).
- 7 Wang, W. *et al.* Measurement of the cleavage energy of graphite. *Nature Communications* **6**, 1-7, doi:10.1038/ncomms8853 (2015).
- 8 Qu, C., Wang, K., Wang, J. & Zheng, Q. Origin of friction in superlubric graphite contacts. *Physical Review Letters* **125**, 126102, doi:10.1103/PhysRevLett.125.126102 (2020).
- 9 Wang, K., Qu, C., Wang, J., Quan, B. & Zheng, Q. Characterization of a Microscale Superlubric Graphite Interface. *Physical Review Letters* **125**, 026101, doi:10.1103/PhysRevLett.125.026101 (2020).
- 10 Berman, D., Deshmukh, S. A., Sankaranarayanan, S. K. R. S., Erdemir, A. & Sumant, A. V. Macroscale superlubricity enabled by graphene nanoscroll formation. *Science* **348**, 1118-1122, doi:10.1126/science.1262024 (2015).
- 11 Kawai, S. *et al.* Superlubricity of graphene nanoribbons on gold surfaces. *Science* **351**, 957-961, doi:10.1126/science.aad3569 (2016).
- 12 Peng, D. *et al.* 100 km wear-free sliding achieved by microscale superlubric graphite/DLC heterojunctions under ambient condition. (Under review)
- 13 Card, H. C. & Rhoderick, E. H. STUDIES OF TUNNEL MOS DIODES .1. INTERFACE EFFECTS IN SILICON SCHOTTKY DIODES. *Journal of Physics D-Applied Physics* **4**, 1589, doi:10.1088/0022-3727/4/10/319 (1971).
- 14 Singh, A., Reinhardt, K. C. & Anderson, W. A. Temperature-Dependence Of The Electrical Characteristics Of Yb/P-Inp Tunnel Metal-Insulator-Semiconductor Junctions. *J. Appl. Phys.* **68**, 3475-3483, doi:10.1063/1.346358 (1990)

REVIEWERS' COMMENTS

Reviewer #1 (Remarks to the Author):

The author well addressed all my previous concerns that I would like to recommend it to be published in Nature Communication as soon as possible. Very nice work.

Reviewer #2 (Remarks to the Author):

The questions have been addressed properly and the quality of the manuscript has been improved significantly. Therefore, I would suggest the acceptance of the manuscript in current version. Congratulations to all the authors.

Reviewer #3 (Remarks to the Author):

The authors very carefully and satisfactorily addressed my previous comments and concerns. I believe the manuscript is at the shape acceptable for the publication.